# Evaluation of Drought Tolerance of Five Maize Genotypes by Virtue of Physiological and Molecular Responses

**Khalil M. Saad-Allah** [1,*] , **Afaf A. Nessem** [1], **Mohsen K. H. Ebrahim** [1,2] **and Dina Gad** [3]

1   Botany Department, Faculty of Science, Tanta University, Tanta 31527, Egypt;
    afaf.mahmoud@science.tanta.edu.eg (A.A.N.); mohsen.ibrahim@science.tanta.edu.eg (M.K.H.E.)
2   Biology Department, Faculty of Applied Sciences, Umm Al-Qura University,
    Makkah Al-Mukarramah 24382, Saudi Arabia
3   Botany and Microbiology Department, Faculty of Science, Menoufia University,
    Shebin EL-Koum 32511, Egypt; dina.gad@science.menofia.edu.eg
*   Correspondence: khalil.saadallah@science.tanta.edu.eg; Tel.: +20-101-377-9385

**Abstract:** Drought has been recognized as a potential challenge to maize production around the world, particularly in arid and semi-arid regions. The primary focus of the present study was to investigate the metabolic and physiological adjustment mechanisms as well as drought-responsive gene expression patterns in five maize (*Zea mays* L.) genotypes (G314, G2, G10, G123, and G326) with varying drought-tolerance capacities at the vegetative stage. Twenty-one days-old maize plants from five maize genotypes were submitted to a well-watered (10 days) watering interval as a control, mild water stress (15 day interval), and severe water stress (20 day interval) treatments in a field experiment for two successive seasons (2019 and 2020). For all maize genotypes, the results showed that water stress significantly reduced plant height, leaf area, biomass, and yield characteristics. However, water stress, which was associated with the length of the watering interval, increased the concentrations of glycine betaine, amino acids, proline, phenols, flavonoids, soluble proteins, and soluble sugars, as well as catalase and peroxidase activities. On the transcriptional level, prolonged water stress increased the expression of drought-responsive genes (LOS5, Rad17, NCED1, CAT1, and ZmP5CS1), with G10 and G123 genotypes being the most drought-resistant. Herein, genotypes G10 and G123 were shown in this study to be relatively water stress tolerant due to improved osmoregulatory, antioxidant, and metabolic activities under water stress conditions, as well as the fact that they were endowed with stress-responsive genes.

**Keywords:** maize genotypes; drought tolerance; oxidative stress; osmoregulation; qRT-PCR

## 1. Introduction

Water deficit is one of the yield-limiting challenges restraining agricultural productivity around the world, particularly in the Mediterranean region. In addition, the climate change crisis and the predictions offered foretell that water supply will be a major predicament for many nations in the coming years [1]. As a result, it is necessary to maximize the utilization of water supplies to secure them. Hence, electing proper plant species with low water demand that are more drought-tolerant is one of the most important policies to conserve water [2]. The typical method for tackling the issue of water stress on field crops was to develop drought-tolerant genotypes. The traditional breeding programs have indeed contributed to the development of high-yielding and drought-tolerant genotypes [3].

Water stress impairs normal plant growth and stimulates biochemical, morphological, and physiological alterations; however, understanding the complex pathways and mechanisms of plant responses to water stress is critical for understanding plant adaptability and endurance during subsequent drought periods [4]. The disruptions in stomatal conductance, water status, photosynthetic activity, assimilates and nutrients flow, leaf temperature, hormonal activity, and osmolytes accumulation, are all signs of water stress

in plants [5]. Recently, Bhusal et al. [6] stated that seedlings of *Prunus sargentii* and *Larix kaempferi* were adversely affected by drought in terms of morphological and physiological features. Drought reduced the size and the growth of branches in both species. It also decreased photosynthesis, electron transfer, stomatal conductance, leaf water potential, carbon isotopic composition, and sap flow into the xylem vessel. Under a drought situation, a biochemical signal is conveyed from roots to leaves via the xylem, causing partial stomatal closure and a substantial decline in internal $CO_2$ concentration [7–9]. As a result, the flow of electrons to the Calvin cycle is inhibited, hindering the electron transport chain and lowering the amount of accessible $NADP^+$, increasing ROS levels [10,11]. Increased levels of ROS cause oxidative damage to biomolecules, predominantly membrane lipids, shifting the plant's metabolic pathways away from growth and biomass synthesis toward stress defense.

Because of their variability in genetically based antioxidant systems, plants respond to environmental stresses differentially, producing tolerant and sensitive genotypes within the same species [12]. Plants modify their physiological processes and hydraulic signals in short-term adaptations; nevertheless, long-term adaptations can influence hundreds of associated gene expression mechanisms and, as a result, phenotypes and possibly genotypes of the plant species [13,14]. Plants predominantly prevent the damage caused by ROS by managing the concentration of osmoregulating compounds and controlling the antioxidant system. The antioxidant system includes both enzymatic (e.g., SOD, POX, CAT, and GR) and non-enzymatic (e.g., phenols, flavonoids, glutathiones, and ascorbate) antioxidants. However, the principal osmoregulating molecules include proline, soluble carbohydrates, and proteins [15], in addition to free amino acids, glycine betaine, and polyamines. Plants may sustain normal growth under oxidative stress when there is equilibrium between ROS and antioxidant systems, but when there is an imbalance between ROS production and antioxidant systems, plant growth is hampered, resulting in significant yield losses.

By global production, maize (*Zea mays* L.) is ranked as the third most important staple food crop in the world. From the Southern to the Northern Hemispheres, and from arid and semi-arid to humid and semi-humid regions, it is grown with an average cultivated area of 157 million ha and yield of 781 Mt between 2000 and 2014 [16]. To fulfill future food security, maize yield must be boosted, particularly in food-insecure countries. However, according to the meta-analysis conducted by Daryanto et al. [17] on maize yield based on 35 years of available data (1980–2015), water stress reduced it by approximately 40% on a worldwide scale. Drought severity, exposure period, and the growth stage are all factors that influence maize yield loss. Drought exposure during the seedling stage causes a substantial reduction in overall maize biomass, and during the jointing to milk stages it alters maize phenotype and decreases its yield. Furthermore, at the same drought intensity, photosynthesis is reduced more at the tasseling stage than at the jointing and milk stages [18]. Furthermore, Song and Jin [19] showed that water stress at the seedling stage caused maize anthesis and maturity dates to be delayed, as well as the growth stage to be prolonged. Also, Çakir [20] concluded that water stress during the vegetative and tasseling stages reduced the total biomass and triggered early loss of lower leaves and poorer grain yield during the ear development and milking stages due to less captured solar energy. On the molecular level, drought stress generates changes in the expression levels of some transcription factors implicated in the regulation of maize root growth and development, allowing some genotypes to be more drought-tolerant [21]. In addition, Zhang et al. [22] demonstrated that the difference in the expression of specific circular RNAs, which varied in their expression in maize under water stress situations, was correlated to drought tolerance. Ren et al. [23] demonstrated that the transcriptional factor ZmNST3 improves maize drought resistance by regulating the expression of genes involved in cell wall formation, antioxidant enzymes, and secondary metabolites.

To improve maize drought tolerance, several agronomic strategies have been implemented, including the use of osmoprotectants, phytohormones, microbial colonizing, soil conditioning, seed priming, and genetic engineering programs. Indeed, assessing local

genotypes' drought-tolerance is essential for considering maize genotypes for successful breeding programs. The objective of this study was to use morphological, physiological, and molecular data to investigate the variations in water-stress tolerance of five maize genotypes, commonly grown in Egypt, under three different water regimes. We also aimed to detect the most tolerant genotype that could be used in water-restricted areas that might be employed as parents for water-stress-tolerant genotypes.

## 2. Materials and Methods

### 2.1. Plant Material and Experimental Design

Seeds of five genotypes of maize (*Zea mays* L.) were obtained from the Agricultural Research Center in Giza, Egypt. The provided genotypes were identified as "Giza 314, Giza 2, Giza 10, Giza 123, and Giza 326". The experiment was conducted at the Research Farm of the Faculty of Agriculture, Sebarbay, Tanta, El-Gharbia, Egypt ($30°82'$ N, $30°99'$ E) during the summer seasons of 2019 and 2020. For both years, the soil at the study site was clayey (13.6% sand, 25.3% silt, and 56.8% clay), with an average pH of 7.78 and electrical conductivity (EC) of 5.49 ds.m$^{-2}$. The average mineral composition of the soil (%) was 1.22 N, 0.72 P, 1.63 Ca, 0.89 K, 0.46 Mg, and 0.62 Na. The research site's weather conditions were semi-arid with no summer rains (rainfall 0.0 mm, average day/night temperature of $38.6/15.7 \pm 2$ °C, and relative humidity of 55%). In each season, there were three replications of a randomized complete block design in a split-plot layout, with a plot area of 12 m$^2$ ($3 \times 4$). The main plot was allocated irrigation interval treatments, whereas maize genotypes were assigned to the sub-plots, with three replications for each. Maize seeds were surface-sterilized for 5 min with 80% aqueous ethanol, thoroughly rinsed three times with tap water, and then once with distilled water prior to sowing. Two maize seeds were sown per hole, with a spacing of 30 cm between holes and 70 cm between rows. Seedlings were thinned to one per hole two weeks after planting at the 2–4 leaves stage according to Karimmojeni et al. [24]. The following watering regimens were used to promote drought stress during the vegetative stage (21 days old): (1) the control, which included watering every 10 days through the growing season, (2) mild water stress, which was generated by watering every 15 days, and (3) severe water stress, created by watering every 20 days. Plots were watered on the planned irrigation days based on 100% field capacity (261 L/m$^2$) during the vegetative stage of growth. Following that, all maize plots were well-watered (100% FC) every 10 days at the start of the reproductive stage until harvest.

### 2.2. Evaluation of Growth Performance

Sixty-five days after the start of the relevant treatments, three plants from each treatment were harvested and divided into shoots and roots. Shoot height (cm/plant), leaf area (cm$^2$/leaf), and shoot biomass (g/plant) were determined in the harvested samples in triplicates. Shoot height was measured using graduate tape and leaf area was measured in the fifth fully expanded leaf of three plants using a BenQ 500B flatbed scanner (BenQ Inc., Taipei, Taiwan), and calculated (cm$^2$) using the Scion imaging software. To calculate shoot biomass, shoot samples were dried for five days at 65 °C in an air-forced oven.

### 2.3. Assessment of Yield Attributes

At the time of harvest, each plot's plants were picked separately, and the cobs were detached from the plant and allowed to dry in the open air. The grains of each treatment in each of the five genotypes were manually extracted from the maize cobs, and the yield parameters were calculated: grain yield (kg/fed) and weight of 100 grains (g).

### 2.4. Osmoregulatory Molecules Determination

Three osmoregulatory compounds were measured in finely powdered dry maize leaves: glycine betaine, free amino acids, and free proline. The colorimetric approach suggested by Grieve and Grattan [25] was used to determine glycine betaine (GB). The data were reported as mg/g DM using a GB standard curve. Powdered leaf samples were

extracted overnight in ethanol (80%) to quantify free amino acids by the approach of Lee and Takahashi [26]. The free amino acids were calculated (mg/g DM) using a standard graph made with glycine. According to Bates et al. [27], free proline in maize genotypes leaf powders was extracted and quantified. By using a standard graph constructed with proline, the concentration of free proline (mg/g DM) was determined.

### 2.5. Evaluation of Phenolics and Flavonoids

The ethanolic extract of dry powdered maize leaves was analyzed for total phenols and flavonoids. Khanam et al.'s [28] technique using Folin-Ciocalteu's reagent was used to quantify total phenolic content. Leaf extracts were combined with Folin-Ciocalteu's reagent and a 20% $Na_2CO_3$ solution, then incubated in the dark for 1 h before reading the absorbance at 760 nm. A standard graph using gallic acid was utilized to determine phenolic compounds in mg/g DM. To assess the flavonoidal content of maize leaf extracts, Khanam et al.'s [28] colorimetric technique was used. Methanol, 10% $AlCl_3$, and 1 M $KCOOCH_3$ were combined with ethanolic leaf extracts. The absorbance of the reaction mixture was spectrophotometrically measured at 415 nm after 30 min. Total flavonoids were represented as mg/g DM, with quercetin standing as the reference flavonoid.

### 2.6. Activities of CAT and POD

Liquid nitrogen was used to pulverize fresh maize leaves, which were then extracted in a 50 mM phosphate buffer containing EDTA and PVP. In a cooling centrifuge, the mixture was centrifuged for 20 min at 10,000 rpm. Catalase and peroxidase assays were performed on the supernatant. The enzyme extract was mixed with 50 mM K-phosphate buffer (pH 7.0), 15 mM $H_2O_2$ was used to initiate the reaction, and the absorbance was measured at 240 nm for assaying catalase [29]. For peroxidase assay, the enzyme extract was combined with 50 mM K-phosphate buffer (pH 5.8), 13 mM guaiacol, and 5 mM $H_2O_2$ to start the reaction, and then the absorbance at 460 nm was measured [29]. The extinction coefficients of 40.0 and 26.6 $mM^{-1}\,cm^{-1}$ were used to calculate the activities of catalase and peroxidase, respectively, as $\mu M/g\,FM.min^{-1}$.

### 2.7. Soluble Proteins and Sugars Determination

Borate buffer (pH 8.0) was used to extract soluble proteins and sugars from maize leaves. Total soluble proteins were measured as mg/g DM according to Bradford [30] using Commassi brilliant blue and standard graph created by bovine serum albumin. The phenol-sulfuric acid method adopted by Dubois et al. [31] was used to quantify total soluble sugars using glucose as a reference sugar.

### 2.8. PCR Analysis

### 2.8.1. Extraction of Total RNA and Synthesis of cDNA

The RNeasy Mini Kit (Qiagen) was used to extract total RNA from maize leaves according to the manufacturer's instructions. Complementary DNA (cDNA) was synthesized through reverse RNA transcription in a total volume of 20 μL using QuantiTect Reverse Transcription Kit and a thermocycler (MJ Research, Inc., PTC-100™ Programmable thermal controller, Waltham, MA, USA). The protocol included a first enzyme activation cycle (42 °C) for 60 min and a second enzyme inactivation cycle (95 °C) for 5 min.

### 2.8.2. qRT PCR Analysis

With the SYBR Green PCR Master Mix (Fermentas, Waltham, MA, USA), the qRT-PCR was carried out in triplicate. In each reaction, a 25 μL mixture containing primer pairs of drought resistance genes (LOS5, Rad17, NCED1, CAT1, or ZmP5CS1) were utilized (Table 1), and data were acquired throughout the extension stage.

**Table 1.** Sequences of primers used in RT-PCR (5′–3′).

| Gene Name | Forward Primer | Reverse Primer |
|-----------|----------------|----------------|
| LOS5 | TGATGCTGCAAAGGGTTGTGCTAC | AATTGAAGCAGCAACAGTGCCTCC |
| Rad17 | CCCATAAGTACAGTGGCTGTGCT | ACGTACAAATTCACCCCACAAGTA |
| Zm-NCED1 | AGTTGTTGTCACCCAGTCCAG | CACGCACCGATAGCCACA |
| CAT1 | CTAACAGGCTGTCGTGAGAAGTG | TGTCAGTGCGTCAACCCATC |
| ZmP5CS1 | ACTGCAATGTCCACTTATCC | TAACCTAGACTAGACACAGC |
| β-actin | GATTCCTGGGATTGCCGAT | TCTGCTGCTGAAAAGTGCTGAG |

A Rotor-Gene 6000 (QIAGEN, ABI System, Zanesville, OH, USA) was used to carry out the reaction. The amplification protocol included a 10 min denaturation step at 95 °C, followed by 40 cycles of 5 s at 95 °C, 30 s at 60 °C, and 30 s at 72 °C. The melting curves were acquired to avoid the implication of non-specific products. As a control gene, the gene β-actin was used [32]. The $2^{-\Delta\Delta Ct}$ method was used to examine gene expression data [33].

*2.9. Statistical Analysis*

All of the experimental results were merged and analyzed over two years, because it was assumed that genotype effects were constant and year effects were random. Minitab 19.11 software was used to analyze and assess the data utilizing analysis of variance (ANOVA) in the general linear model (GLM). Fixed factors included five maize genotypes and three watering intervals, whereas dependent variables comprised all of the morphological and physiological parameters studied. In addition, if significant differences were observed, a posthoc test was performed. As a posthoc test, Tukey's HSD test at the 5% level was employed to evaluate the significant differences in the interaction effects.

One-way analysis of variance (One-way ANOVA) and Tukey's HSD were used to determine the differences in gene expression between water stress treated and control maize genotypes. A statistically significant *p*-value was defined as less than 0.05. The results are reported as the average of two separate studies with six replicates (3 each).

**3. Results**

*3.1. Analyses of Variance of Genotypes, Drought Intervals, and Their Interaction Effects on the Agronomic Traits of Maize in a Two-Field Experiment Conducted in 2019 and 2020*

Table 2 presents the results of a two-way completely randomized analysis of variance (ANOVA) on the studied maize parameters for the impacts of irrigation intervals, maize genotypes, and their combined interaction. According to the results, drought intervals and maize genotypes have a highly significant (*p* < 0.001) effect on all of the studied attributes. Concerning irrigation intervals and maize genotypes interactions, there were highly significant interactions for plant height, leaf area, amino acids, proline, catalase, and peroxidase. However, these interactions were significant (*p* < 0.01) for biomass, yield/fed, glycine betaine, phenolics, flavonoids, soluble proteins and soluble sugars, but these interactions were significant (*p* < 0.05) for the weight of 100 grains.

**Table 2.** Results of two-way ANOVA analysis.

| Parameters | Genotype | Drought | Genotype × Drought |
|---|---|---|---|
| Plant height (cm) | 45.20 *** | 1393.26 *** | 9.18 *** |
| Leaf area (cm$^2$) | 151.41 *** | 1126.38 *** | 5.42 *** |
| Biomass (g/plant) | 9.12 *** | 94.87 *** | 2.08 ** |
| Grain yield (kg/fed) | 43.66 *** | 678.23 *** | 3.67 ** |
| Weight of 100 grains (g) | 13.30 *** | 143.34 *** | 4.35 ** |
| GB (mg/g DM) | 148.04 *** | 14.96 *** | 3.38 ** |
| Free amino acids (mg/g DM) | 272.67 *** | 139.75 *** | 6.16 *** |
| Free proline (mg/g DM) | 137.27 *** | 124.41 *** | 5.95 *** |
| Phenolics (mg/g DM) | 40.96 *** | 371.43 *** | 4.48 ** |
| Flavonoids (mg/g DM) | 35.77 *** | 595.26 *** | 3.88 ** |
| CAT activity ($\mu$M/g FM.min$^{-1}$) | 212.98 *** | 37.15 *** | 34.18 *** |
| POD activity ($\mu$M/g FM.min$^{-1}$) | 146.53 *** | 128.93 *** | 26.99 *** |
| Soluble proteins (mg/g DM) | 57.49 *** | 107.16 *** | 7.38 ** |
| Soluble sugars (mg/g DM) | 45.90 *** | 71.47 *** | 3.48 ** |

Numbers represent F-values at 0.05 level; ** $p < 0.01$; *** $p < 0.001$; ns, non-significant.

### 3.2. The Effect of Irrigation Intervals on Maize Genotypes Growth

Table 3 shows the response of five maize genotypes typically grown in Egypt; namely Giza 314 (G314), Giza 2 (G2), Giza 10 (G10), Giza 123 (G123), and Giza 326 (G326), to various irrigation intervals (10, 15, and 20 days). Under control irrigation treatment (10 days), both genotypes G10 and G123 showed the highest values of shoot height, leaf area, and biomass production; however, genotype G326 showed the lowest values for the preceding parameters. Genotypes G314 and G2 revealed intermediate values for the growth parameters under the control irrigation interval. The length of irrigation intervals has a significant impact on growth metrics. The growth parameters of all maize genotypes were substantially reduced when the irrigation interval was increased, with the 20 day interval being the harshest for all maize genotypes. The genotypes G314, G2, G10, G123, and G326 experienced shoot height declines of 52.25, 49.02, 56.24, 55.45, and 52.98%, respectively, following this interval compared to the control one. Concerning leaf area, the aforementioned genotypes exhibited declines of 45.42, 41.79, 34.16, 34.73, and 44.61%, respectively, as compared to the control irrigation treatment. Furthermore, as a result of the prolonged irrigation period (20 days), plant biomass was reduced by 31.18, 32.24, 31.63, 30.05, and 29.81% for the aforementioned genotypes compared to the control irrigation treatment (10 days).

**Table 3.** Growth criteria of five maize genotypes as affected by normal (10 days), mild (15 days), and severe (20 days) water stress at the vegetative stage. Means within the same column distinguished by different letters differ significantly according to the Tukey's posthoc HSD test at $p = 0.05$.

| | Shoot Height (cm) | Leaf Area (cm$^2$) | Plant Biomass (g/plant) |
|---|---|---|---|
| **Irrigation Interval** | | | |
| 10 days | 194.94 [A] | 327.76 [A] | 18.38 [A] |
| 15 days | 99.67 [B] | 242.70 [B] | 14.10 [B] |
| 20 days | 90.81 [C] | 197.91 [C] | 12.70 [C] |
| **Genotypes** | | | |

**Table 3.** *Cont.*

|  | Shoot Height (cm) | Leaf Area (cm$^2$) | Plant Biomass (g/plant) |
|---|---|---|---|
| Giza 314 | 122.578 [B] | 236.72 [B] | 15.24 [A] |
| Giza 2 | 123.56 [B] | 231.65 [B] | 14.76 [B] |
| Giza 10 | 143.28 [A] | 289.11 [A] | 16.11 [A] |
| Giza 123 | 141.22 [A] | 291.37 [A] | 16.01 [A] |
| Giza 326 | 111.73 [C] | 231.77 [B] | 13.19 [B] |
| **Irrigation Interval × Genotype** |  |  |  |
| 10 days × Giza 314 | 184.47 [B] | 307.95 [B] | 18.64A [B] |
| 10 days × Giza 2 | 178.07 [B] | 302.18 [B] | 18.30A [B] |
| 10 days × Giza 10 | 225.33 [A] | 360.42 [A] | 19.56 [A] |
| 10 days × Giza 123 | 217.31 [A] | 368.55 [A] | 19.30 [A] |
| 10 days × Giza 326 | 169.51 [B] | 299.71 [B] | 16.13 [ABC] |
| 15 days × Giza 314 | 95.15 [CDEF] | 234.10 [EF] | 14.29 [CDE] |
| 15 days × Giza 2 | 101.80 [CDE] | 216.90 [F] | 13.53 [CDE] |
| 15 days × Giza 10 | 105.89 [CD] | 269.57 [C] | 15.33 [BCD] |
| 15 days × Giza 123 | 109.58 [C] | 263.31 [CD] | 15.20 [BCD] |
| 15 days × Giza 326 | 85.95 [EF] | 229.61 [EF] | 12.17 [DE] |
| 20 days × Giza 314 | 88.11 [DEF] | 168.12 [G] | 12.81 [CDE] |
| 20 days × Giza 2 | 90.80 [DEF] | 175.87 [G] | 12.43 [DE] |
| 20 days × Giza 10 | 98.62 [CDE] | 237.33 [EF] | 13.45 [CDE] |
| 20 days × Giza 123 | 96.78C [DEF] | 242.25 [DE] | 13.54 [CDE] |
| 20 days × Giza 326 | 79.73 [F] | 166.00 [G] | 11.28 [E] |

*3.3. The Effect of Irrigation Intervals on Maize Genotypes Yield Parameters*

Drought exposure, as recognized by irrigation intervals in the current study, had a significant impact on the studied maize genotypes yield metrics (Table 4). Genotypes G10 and G123 had the highest grain yields under control irrigation treatment (10 days), with grain yields of 3248.4 and 3246.0 kg/fed, respectively. The G326 genotype had the lowest grain production, whereas G314 and G2 genotypes recorded midrange values of grain yield under control irrigation treatment. The rate of grain yield loss was dependent on the length of the watering interval. The 20 day irrigation interval resulted in a significant decrease in maize yield. The grain yield reduction following this irrigation interval for G314, G2, G10, G123, and G326 genotypes were 18.35, 18.05, 14.58, 17.00, and 18.28%, respectively. As a result, when compared to other genotypes, G10 and G123 demonstrated better drought tolerance in terms of grain yield.

**Table 4.** Yield parameters of five maize genotypes as affected by normal (10 days), mild (15 days), and severe (20 days) water stress at the vegetative stage. Means within the same column distinguished by different letters differ significantly according to the Tukey's posthoc HSD test at *p* = 0.05.

|  | Grain Yield (kg/fed) | Weight of 100 Grains (g) |
|---|---|---|
| **Irrigation Interval** |  |  |
| 10 days | 3172.46 [A] | 30.25 [A] |
| 15 days | 2906.13 [B] | 25.48 [B] |
| 20 days | 2625.33 [C] | 23.17 [C] |
| **Genotypes** |  |  |

**Table 4.** *Cont.*

|  | Grain Yield (kg/fed) | Weight of 100 Grains (g) |
|---|---|---|
| Giza 314 | 2889.05 [B] | 24.85 [B] |
| Giza 2 | 2851.98 [B] | 25.78 [B] |
| Giza 10 | 2999.21 [A] | 27.99 [A] |
| Giza 123 | 2981.38 [A] | 28.11 [A] |
| Giza 326 | 2784.92 [C] | 24.79 [B] |
| **Irrigation Interval ×** **Genotype** | | |
| 10 days × Giza 314 | 3177.81 [AB] | 30.43 [AB] |
| 10 days × Giza 2 | 3139.95 [BC] | 29.75 [AB] |
| 10 days × Giza 10 | 3248.31 [AB] | 31.43 [A] |
| 10 days × Giza 123 | 3276.13 [A] | 31.52 [A] |
| 10 days × Giza 326 | 3020.08 [CD] | 28.12 [ABC] |
| 15 days × Giza 314 | 2897.33 [EF] | 24.05 [DEF] |
| 15 days × Giza 2 | 2842.67 [FG] | 24.56 [CDEF] |
| 15 days × Giza 10 | 2974.67 [DE] | 26.87 [BCD] |
| 15 days × Giza 123 | 2949.33 [DEF] | 27.16 [BCD] |
| 15 days × Giza 326 | 2866.67 [EFG] | 24.75 [CDEF] |
| 20 days × Giza 314 | 2592.00 [I] | 20.08 [G] |
| 20 days × Giza 2 | 2573.33 [IJ] | 23.02 [EFG] |
| 20 days × Giza 10 | 2774.67 [GH] | 25.67 [CDE] |
| 20 days × Giza 123 | 2718.67 [H] | 25.62 [CDE] |
| 20 days × Giza 326 | 2468.00 [J] | 21.50 [FG] |

In different maize genotypes, the grain weight as a yield quality criterion was affected by the watering interval (Table 4). Genotypes G10 and G123 recorded the highest value of 100-grain weight (31.5 g) with the control watering treatment (10 days), whereas the G326 genotype had the lowest value (28.0 g). The weight loss of 100 grains was directly proportional to the length of the irrigation period, with a 20 day irrigation interval significantly reducing maize grain weight in all studied genotypes. For genotypes G314, G2, G10, G123, and G326, the reductions in 100-grain weight attained by this irrigation interval were 34.43, 19.46, 18.10, 18.09, and 23.21%, respectively. As a result, G10 and G123 were found to be the most drought-tolerant genotypes in this study based on yield characteristics.

*3.4. The Effect of Irrigation Intervals on Maize Genotypes Osmoregulatory Molecules*

Within maize genotypes, watering intervals had a considerable impact on osmoregulatory components such glycine betaine (GB), free amino acids, and proline (Table 5). Under control watering treatment (10 days), the amounts of GB, amino acids, and proline reached their highest levels in the G123 and G10 genotypes. Under the same conditions, the G314 genotype had the lowest GB level, whereas the G326 genotype had the lowest amino acids and proline levels.

**Table 5.** Osmoregulators of five maize genotypes as affected by normal (10 days), mild (15 days), and severe (20 days) water stress at the vegetative stage. Means within the same column distinguished by different letters differ significantly according to the Tukey's posthoc HSD test at *p* = 0.05.

| | GB (mg/g DM) | Free Amino Acids (mg/g DM) | Free Proline (mg/g DM) |
|---|---|---|---|
| **Irrigation Interval** | | | |
| 10 days | 3.38 [B] | 26.65 [C] | 18.87 [C] |
| 15 days | 3.62 [A] | 30.67 [B] | 25.63 [B] |
| 20 days | 3.71 [A] | 32.64 [A] | 28.36 [A] |
| **Genotypes** | | | |
| Giza 314 | 2.95 [D] | 27.21 [B] | 20.84 [B] |
| Giza 2 | 3.13 [CD] | 28.15 [B] | 20.32 [B] |
| Giza 10 | 3.98 [B] | 35.87 [A] | 30.82 [A] |
| Giza 123 | 4.57 [A] | 35.47 [A] | 31.98 [A] |
| Giza 326 | 3.24 [C] | 23.25 [C] | 17.48 [C] |
| **Irrigation Interval × Genotype** | | | |
| 10 days × Giza 314 | 2.66 [F] | 24.90 [EFG] | 15.32 [DE] |
| 10 days × Giza 2 | 3.05 [EF] | 23.27 [G] | 16.31 [DE] |
| 10 days × Giza 10 | 3.74 [CD] | 29.82 [CD] | 25.83 [B] |
| 10 days × Giza 123 | 4.40 [B] | 33.81 [BC] | 25.54 [B] |
| 10 days × Giza 326 | 3.08 [EF] | 21.47 [G] | 11.34 [E] |
| 15 days × Giza 314 | 3.01 [EF] | 27.96 [DEF] | 22.17 [BC] |
| 15 days × Giza 2 | 3.12 [EF] | 30.28 [CD] | 21.52 [BC] |
| 15 days × Giza 10 | 4.12 [BC] | 36.35 [B] | 31.64 [A] |
| 15 days × Giza 123 | 4.58 [AB] | 35.14 [B] | 34.51 [A] |
| 15 days × Giza 326 | 3.30 [DE] | 23.66 [G] | 18.34 [CD] |
| 20 days × Giza 314 | 3.18 [E] | 28.77 [DE] | 25.03 [B] |
| 20 days × Giza 2 | 3.23 [E] | 30.91 [CD] | 23.13 [BC] |
| 20 days × Giza 10 | 4.10 [BC] | 41.45 [A] | 35.00 [A] |
| 20 days × Giza 123 | 4.72 [A] | 37.46 [B] | 35.91 [A] |
| 20 days × Giza 326 | 3.34 [DE] | 24.61F [G] | 22.76 [BC] |

The accumulation of osmoregulatory molecules in the leaves of maize genotypes was substantially boosted by increasing watering interval. When compared to the control treatment, the 20 day watering interval resulted in 19.55, 5.56, 9.10, 7.27, and 8.09% increases in GB accumulation in G314, G2, G10, G123, and G326 maize genotypes, respectively. Furthermore, when compared to the control irrigation treatment (10 days), this watering interval resulted in 15.54, 32.77, 39.00, 10.789, and 14.62% increases in free amino acid accumulation in the aforementioned maize genotypes, respectively. Within the maize genotypes investigated, the length of irrigation interval had a direct impact on the production of free proline, a stress amino acid. The longest irrigation interval resulted in proline content increases of 63.38, 41.81, 35.50, 40.60, and 100.79% in the G314, G2, G10, G123, and G326 maize genotypes, respectively. Herein, the findings suggest that maize plants can withstand prolonged droughts through accumulating greater amounts of osmoregulators, particularly proline.

*3.5. The Effect of Irrigation Intervals on Maize Genotypes Non-Enzymatic Antioxidants*

Longer watering intervals boosted non-enzymatic antioxidant compounds such phenols and flavonoids in all studied maize genotypes leaves (Table 6). Under control irrigation treatment conditions (10 days), the genotype G10 had the highest phenolic and flavonoidal content (25.25 and 12.08 mg/g DM, respectively), followed by the genotype G123 (25.74 and 11.87 mg/g DM, respectively). In all maize genotypes, the 20 day irrigation interval stimulated the highest levels of phenols and flavonoids. In G314, G2, G10, G123, and G326 genotypes, the increase in phenolic content because of this watering interval was

60.62, 66.10, 71.10, 72.26, and 71.10%, respectively. The rise in flavonoidal content caused by this watering interval was substantially greater than the increase in phenolic content. It increased the flavonoidal content of the aforementioned maize genotypes by 132.00, 141.34, 131.62, 157.71, and 137.97%, respectively. Accordingly, under extreme drought conditions, the G123 genotype accumulated the most phenols and flavonoids among all maize genotypes.

**Table 6.** Non-enzymatic antioxidants of five maize genotypes as affected by normal (10 days), mild (15 days), and severe (20 days) water stress at the vegetative stage. Means within the same column distinguished by different letters differ significantly according to the Tukey's posthoc HSD test at $p$ = 0.05.

| | Phenolics (mg/g DM) | Flavonoids (mg/g DM) |
|---|---|---|
| **Irrigation Interval** | | |
| 10 days | 23.59 [B] | 10.83 [B] |
| 15 days | 38.32 [A] | 24.84 [A] |
| 20 days | 39.72 [A] | 26.05 [A] |
| **Genotypes** | | |
| Giza 314 | 31.53 [BC] | 17.06 [C] |
| Giza 2 | 32.62 [B] | 19.52 [B] |
| Giza 10 | 38.01 [A] | 22.91 [A] |
| Giza 123 | 37.76 [A] | 23.60 [A] |
| Giza 326 | 29.47 [C] | 19.78 [B] |
| **Irrigation Interval × Genotype** | | |
| 10 days × Giza 314 | 22.55 [DE] | 9.50 [F] |
| 10 days × Giza 2 | 22.89 [DE] | 10.28 [F] |
| 10 days × Giza 10 | 26.25 [D] | 12.08 [F] |
| 10 days × Giza 123 | 25.74 [DE] | 11.87 [F] |
| 10 days × Giza 326 | 20.53 [E] | 10.43 [F] |
| 15 days × Giza 314 | 35.83 [C] | 19.64 [E] |
| 15 days × Giza 2 | 36.95 [C] | 23.47 [DE] |
| 15 days × Giza 10 | 42.86 [AB] | 28.68 [AB] |
| 15 days × Giza 123 | 43.22 [AB] | 28.34 [AB] |
| 15 days × Giza 326 | 32.76 [C] | 24.10 [CD] |
| 20 days × Giza 314 | 36.22 [C] | 22.04 [DE] |
| 20 days × Giza 2 | 38.02 [BC] | 24.81 [BCD] |
| 20 days × Giza 10 | 44.91 [A] | 27.98 [ABC] |
| 20 days × Giza 123 | 44.34 [A] | 30.59 [A] |
| 20 days × Giza 326 | 35.11 [C] | 24.82 [BCD] |

*3.6. The Effect of Irrigation Intervals on Maize Genotypes CAT and POD Activities*

In response to maize genotype and drought severity, the activities of two antioxidant enzymes, CAT and POD, were differentially affected (Table 7). Under control irrigation conditions (10 days), genotypes G10 and G123 had the highest CAT activity (1.38 and 1.24 μM/g FM.min$^{-1}$, respectively), whereas genotype G326 had the lowest (0.53 μM/g FM.min$^{-1}$). All maize genotypes showed enhanced CAT activity at the moderate water stress (15 days), whereas the severe water stress (20 days) had a varied effect on maize genotypes' CAT activity. The 20 day interval reduced CAT activity in the G2 and G326 genotypes by 40.86 and 28.30%, respectively, but did not affect the G314 genotype. The G10 and G123 maize genotypes, in contrast, increased their CAT activity by 69.56 and 128.23%, respectively, as a result of the longest watering interval.

**Table 7.** Antioxidant enzymes of five maize genotypes as affected by normal (10 days), mild (15 days), and severe (20 days) water stress at the vegetative stage. Means within the same column distinguished by different letters differ significantly according to the Tukey's posthoc HSD test at $p = 0.05$.

| | CAT Activity ($\mu$M/g FM.min$^{-1}$) | POD Activity ($\mu$M/g FM.min$^{-1}$) |
|---|---|---|
| **Irrigation Interval** | | |
| 10 days | 0.97 B | 106.10 C |
| 15 days | 1.28 A | 112.46 B |
| 20 days | 1.37 A | 135.88 A |
| **Genotypes** | | |
| Giza 314 | 0.85 B | 103.28 B |
| Giza 2 | 0.87 B | 103.14 B |
| Giza 10 | 1.88 A | 142.65 A |
| Giza 123 | 1.92 A | 140.91 A |
| Giza 326 | 0.50 C | 100.61 B |
| **Irrigation Interval $\times$ Genotype** | | |
| 10 days $\times$ Giza 314 | 0.74 GHI | 97.77 E |
| 10 days $\times$ Giza 2 | 0.93 FGH | 96.45 E |
| 10 days $\times$ Giza 10 | 1.38 DE | 123.25 BC |
| 10 days $\times$ Giza 123 | 1.24 EF | 117.48 BCD |
| 10 days $\times$ Giza 326 | 0.53 HI | 95.10 E |
| 15 days $\times$ Giza 314 | 1.06 EFG | 104.91 DE |
| 15 days $\times$ Giza 2 | 1.14 EFG | 104.79 DE |
| 15 days $\times$ Giza 10 | 1.92 C | 125.82 B |
| 15 days $\times$ Giza 123 | 1.68 CD | 124.54 B |
| 15 days $\times$ Giza 326 | 0.60 HI | 102.23 DE |
| 20 days $\times$ Giza 314 | 0.74 GHI | 107.16 CDE |
| 20 days $\times$ Giza 2 | 0.54 HI | 108.18 CDE |
| 20 days $\times$ Giza 10 | 2.34 B | 178.88 A |
| 20 days $\times$ Giza 123 | 2.83 A | 180.70 A |
| 20 days $\times$ Giza 326 | 0.38 I | 104.50 DE |

The G10 and G123 genotypes had the highest POD activity at the control irrigation treatment (123.3 and 117.3 $\mu$M/g FM.min$^{-1}$, respectively). In all maize genotypes, increasing the irrigation interval significantly boosted POD activity. The POD activity of the G314, G2, G10, G123, and G326 genotypes increased by 9.61, 12.24, 45.10, 53.79, and 9.88%, respectively, after a 20 day irrigation interval. As a result, in G10 and G123 genotypes, POD is a fundamental determinant of drought tolerance.

*3.7. The Effect of Irrigation Intervals on Maize Genotypes Protein and Sugar Contents*

The intensity of drought stress (irrigation interval) had a substantial impact on soluble protein and sugar contents of the studied maize genotypes (Table 8). Under control irrigation treatment, G10 and G123 genotypes had the highest protein content in maize leaves (6.21 and 6.37 mg/g DM, respectively). Drought intensity increased total soluble protein content, with the 20 day irrigation interval recording the maximum protein content throughout all maize genotypes. The protein content of G314, G2, G10, G123, and G326 genotypes increased by 23.42, 53.68, 41.38, 44.74, and 38.34%, respectively. Total soluble sugars, like total soluble proteins, peaked in genotypes G10 and G123 (5.00 and 4.92 mg/g DM, respectively). When compared to the control irrigation treatment, this irrigation interval resulted in 50.15, 60.80, 24.60, 41.46, and 52.99% increases in total soluble sugars content of G314, G2, G10, G123, and G326, respectively.

**Table 8.** Soluble proteins and sugars of five maize genotypes as affected by normal (10 days), mild (15 days), and severe (20 days) water stress at the vegetative stage. Means within the same column distinguished by different letters differ significantly according to the Tukey's posthoc HSD test at $p = 0.05$.

| | Soluble Proteins (mg/g DM) | Soluble Sugars (mg/g DM) |
|---|---|---|
| **Irrigation Interval** | | |
| 10 days | 5.45 [C] | 3.92 [C] |
| 15 days | 7.16 [B] | 5.63 [A] |
| 20 days | 7.70 [A] | 5.06 [B] |
| **Genotypes** | | |
| Giza 314 | 6.24 [B] | 4.19 [B] |
| Giza 2 | 6.21 [B] | 4.29 [B] |
| Giza 10 | 7.83 [A] | 5.63 [A] |
| Giza 123 | 8.05 [A] | 6.05 [A] |
| Giza 326 | 5.51 [C] | 4.18 [B] |
| **Irrigation Interval × Genotype** | | |
| 10 days × Giza 314 | 5.33 [EF] | 3.27 [EF] |
| 10 days × Giza 2 | 4.75 [F] | 3.24 [EF] |
| 10 days × Giza 10 | 6.21 [CDE] | 5.00 [CD] |
| 10 days × Giza 123 | 6.37 [CDE] | 4.92 [CD] |
| 10 days × Giza 326 | 4.59 [F] | 3.17 [F] |
| 15 days × Giza 314 | 6.54 [CDE] | 4.91 [CD] |
| 15 days × Giza 2 | 6.60 [CDE] | 5.21 [BCD] |
| 15 days × Giza 10 | 8.52 [AB] | 6.23 [AB] |
| 15 days × Giza 123 | 8.56 [AB] | 6.96 [A] |
| 15 days × Giza 326 | 5.59 [DEF] | 4.85 [CD] |
| 20 days × Giza 314 | 6.85 [CD] | 4.39 [DE] |
| 20 days × Giza 2 | 7.30 [BC] | 4.42 [DE] |
| 20 days × Giza 10 | 8.78 [A] | 5.68 [BC] |
| 20 days × Giza 123 | 9.22 [A] | 6.27 [AB] |
| 20 days × Giza 326 | 6.35 [CDE] | 4.51 [CD] |

*3.8. RT PCR Analysis of LOS5, Rad17, NCED1, CAT1, and ZmP5CS1 Genes*

The expression of key genes that contribute to the plant's tolerance to water stress is linked to the phenology and physiology of maize genotypes exposed to harsh water stress during the vegetative stage. Prolonged irrigation interval (20 days) at the vegetative stage significantly affected the expression levels of some genes related to water stress tolerance in maize, such as molybdenum cofactor sulfurase (LOS5), checkpoint clamp loader component (Rad17), 9-cis-epoxycarotenoid dioxygenase (NCED1), cationic amino acid transporter 1 (CAT1), and delta1-pyrroline-5-carboxylate synthase 1 (ZmP5CS1) (Table 9). The expression of drought resistance genes was used as a molecular marker in this investigation. In general, drought resistance was higher in G10 and G123 maize genotypes compared to G314, G2, and G326 genotypes. Molybdenum cofactor sulfurase (LOS5) revealed greater overexpression levels (1.9 and 1.7 fold, respectively) for the maize genotypes G10 and G123, whereas the least expression value (0.8 fold) was observed with the G2 genotype. Furthermore, the Rad17 gene was expressed 0.9 fold in the G10 and G123 maize genotypes, compared to 0.5 fold in the G2 and G326 maize genotypes. When compared to the G2 genotype (0.4 fold), the NCED1 gene was overexpressed by 1.6 and 1.5 fold in G123 and G10 maize genotypes, respectively. In addition, the CAT1 drought resistance gene was overexpressed by 0.9 and 0.8 fold in the G10 and G123 maize genotypes, respectively, compared to the least expressed genotype (G2), which was only expressed by 0.3 fold. Furthermore, the G10 and G123 maize genotypes overexpressed ZmP5CS1 by 4.0 and 3.2 fold, respectively, compared to the least expressing genotype, G2, which only expressed this gene by 1.9 fold.

Finally, varied responses to water stress were seen in the five maize genotypes. The G10 and G123 genotypes were shown to be the most drought-resistant when compared to the G2 genotype, which exhibited the lowest drought resistance patterns on a molecular level.

**Table 9.** Gene expression of LOS5, Rad17, NCED1, CAT1, and ZmP5CS1 stress-responsive genes in five maize genotypes subjected to severe water stress (20 days irrigation interval) at the vegetative stage. Values are means of three replicates $\pm$ SD. Different letters within the same row donate significant differences among genotypes based on Tukey's posthoc HSD test at $p = 0.05$).

| Gene Name | Giza 314 | Giza 2 | Giza 10 | Giza 123 | Giza 326 |
|:---:|:---:|:---:|:---:|:---:|:---:|
| **LOS5** | 1.2 $\pm$ 0.10 [B] | 0.8 $\pm$ 0.05 [C] | 1.9 $\pm$ 0.04 [A] | 1.7 $\pm$ 0.08 [A] | 1.1 $\pm$ 0.10 [BC] |
| **Rad17** | 0.7 $\pm$ 0.05 [B] | 0.5 $\pm$ 0.02 [C] | 0.9 $\pm$ 0.05 [A] | 0.9 $\pm$ 0.02 [A] | 0.5 $\pm$ 0.03 [C] |
| **NCED1** | 1.1 $\pm$ 0.08 [B] | 0.4 $\pm$ 0.01 [D] | 1.5 $\pm$ 0.12 [A] | 1.6 $\pm$ 0.08 [A] | 0.8 $\pm$ 0.01 [C] |
| **CAT1** | 0.6 $\pm$ 0.06 [B] | 0.3 $\pm$ 0.00 [C] | 0.9 $\pm$ 0.01 [A] | 0.8 $\pm$ 0.06 [A] | 0.4 $\pm$ 0.04 [C] |
| **ZmP5CS1** | 3.1 $\pm$ 0.11 [B] | 1.9 $\pm$ 0.03 [D] | 4.0 $\pm$ 0.15 [A] | 3.2 $\pm$ 0.14 [B] | 2.3 $\pm$ 0.17 [C] |

## 4. Discussion

The current investigation demonstrated that water stress impaired the growth and productivity of all maize genotypes, as well as their physiological and biochemical performance. Water deficit conditions reduced all maize genotypes' growth parameters (shoot height, leaf area, and biomass) significantly, with the 20 day irrigation interval during the vegetative stage being the most detrimental. The reduction in maize growth attributes caused by water stress could be attributed to constraints in stomatal performance and root structure, which obstruct the flow of $CO_2$, water, and nutrients for normal metabolic activities [34]. Also, water stress has been shown to impair cellular division and proliferation by limiting the activity of key genes like tubulin and cyclin [35]. According to Ali et al. [36], intense dryness restricted maize root growth by limiting root penetration through dry soil and lowering root respiratory performance. According to Avramova et al. [37], the fundamental cause of maize growth restriction due to water stress is the substantial reduction in cell division levels in the meristem caused by down-regulation of the complete cell cycle machinery and up-regulation of a cell cycle inhibitor (krp2).

Increased drought intensity (irrigation interval) lowered grain production (kg/fed) and grains weight (100 grain weight) in all maize genotypes, with G314 and G326 genotypes being the most affected. Water stress affects maize production in a variety of ways, including plant developmental stage, the severity and length of water stress, genotype vulnerability, and soil water-stress sensitivity [38]. According to Ge et al. [39], water stress caused a considerable impairment in ear development and poor grain filling, resulting in considerably fewer seeds in each ear. Prolonged and intensified soil water stress may result in the early loss of older leaves, as well as decreased biomass production and yield components, as a result of limited light absorption [20]. Moreover, maize yield may be lowered due to tasseling and silking desynchronization, resulting in delayed grain setting and kernel abortion, depending on the duration and magnitude of drought stress [40]. Furthermore, under inadequate irrigation conditions, maize growth and productivity may be reduced due to inhibited cell division and expansion [41].

All maize genotypes selected for this study accumulated reasonable amounts of the osmoregulatory components glycine betaine (GB), free amino acids, and proline in response to water stress, with the variability of their amounts according to the irrigation interval and the genotype. Many plant species have been shown to utilize GB as the key stress-induced molecule participating in osmoregulation and cellular structure maintenance [42]. Interestingly, GB accumulates in the chloroplast, whereas proline accumulates in the cytoplasm, implying that their osmoregulation roles are complementary. Proline, however, accumulates more quickly than GB. As a result, proline is the first osmoregulatory molecule to accumulate during the initial hours of drought exposure, whereas GB accumulates if the drought impact is prolonged [43]. The low amounts of GB seen in this study are indeed explained by Luo et al. [44], who suggested that the peculiar post-transcriptional

activation of the betaine aldehyde dehydrogenase and choline monooxygenase genes may help understand the low concentrations of GB reported in some cereals. Other precursors, such as abscisic acid, may also be synthesized in response to a stressor to promote the transcription of genes encoding enzymes involved in GB biosynthesis [43]. GB improves PSII repair via sustaining the repair machinery instead of directly defending PSII from photodamage, allowing for effectual photosynthesis under drought situations [45]. Increased GB concentrations have been linked to the activation of novel stress-related genes, the sequestration of reactive oxygen species, and a stabilizing agent for the photosynthetic apparatus and protein structure maintenance by serving as a molecular chaperone during stressful conditions [46]. Besides protecting and preserving enzymes and other macromolecules, proline functions as an osmolyte, offering a defense mechanism against low water potential situations through osmotic adjustment and as an electron acceptor, avoiding photosystem damage by neutralizing ROS [47]. Higher proline concentration causes cytosolic acidosis while also sustaining the $NADP^+/NAD^+$ balance and function as a sink for excess oxidizing agents at the same time [48]. Drought exposure also increased the amount of free amino acids in all maize genotypes, possibly due to increased activity of proteases or de novo enhanced biosynthesis of free amino acids, which serve as a nitrogen reservoir and osmotica [48]. As a result, it can be concluded that the accumulation of GB, proline, and free amino acids in maize reduces cell osmotic potential and minimizes water loss under water stress, particularly in the G10 and G123 genotypes.

Plants can produce more phenols whenever they are exposed to environmental stress. The same biosynthetic pathway of phenolics and flavonoids, as well as the fact that both compounds belonging to a large family named polyphenols, could explain the positive relationship [49]. Under normal and water-stressed conditions, the G10 and G314 genotypes had the highest levels of total phenols and flavonoids, relative to the other genotypes. Drought stress resulted in an increase in total phenols and total flavonoids across all genotypes, with the maximum value of their content in all maize genotypes at the longest watering interval (20 days). Total phenols and flavonoids were found to be closely related to maize drought resistance potential, according to the findings of this study. Plants' reactions to drought stress have been linked to the accumulation of phenolic acids and flavonoids as antioxidants. According to Cappellari et al. [50], increased activity of phenylalanine ammonia-lyase, a key enzyme in phenylpropanoid metabolism, triggers the formation of trans-cinnamic acid, which yields phenolics and flavonoids. Polyphenols can directly bind transition metals, effectively detoxify molecular species of oxygen free radicals, and trap the lipid alkoxyl radicals to prevent lipid peroxidation. Furthermore, peroxidase oxidizes flavonoids and phenylpropanoids, which operate as $H_2O_2$-scavengers in the phenolic/ascorbate/peroxidase mechanism [51]. According to Ayaz et al. [52], water stress triggers lignification of cell walls as well as the biosynthesis of amino acids (predominately phenylalanine and tyrosine) to sustain cellular osmotic balance.

Plants with more potent antioxidant systems, enzymatic and non-enzymatic, are more resistant to stress. As a result, drought tolerance may be dependent on boosting the endogenous antioxidant system, which comprises antioxidant molecules and antioxidant enzymes. Moderate water stress (15 day irrigation interval) during the vegetative phase increased catalase (CAT) activity, whereas severe water stress (20 day irrigation interval) lowered it in all maize genotypes tested, with the maximum activity documented in G10 and G123 genotypes. Drought-induced increases in CAT activity was reported to improve membrane integrity and $CO_2$ fixation through lowering $H_2O_2$ levels [53]. Also, the increased activity of CAT most likely allowed for the elimination of photorespiratory $H_2O_2$ generated in water-stressed plants [54]. The decreased activity of CAT under severe water stress is thought to be due to an impairment of enzyme synthesis or an alteration in the assembly of enzyme subunits. It could potentially be due to photo-inactivation of the enzyme or degradation caused by increased peroxisomal proteases [55]. Nonetheless, when maize genotypes were exposed to water stress, peroxidase (POD) activity increased in all of them, with the G10 and G123 genotypes having the highest activity among the

others. The increased POD activity, especially under severe water stress, was believed to compensate for the reduction of $H_2O_2$ scavenging capability caused by the decline in CAT activity [56]. As it is well-known that plants use basic physiological systems to combat the negative effects of ROS by maintaining high antioxidant levels, POD activity was shown to be higher than CAT activity in all maize genotypes, implying that it is the main $H_2O_2$ scavenging enzyme.

The osmotic adjustment to water deficit is known to be associated with soluble proteins and sugars. Our findings showed that increasing the irrigation interval increased the amount of soluble proteins and sugars in all maize genotypes, with the G123 and G10 genotypes having the highest levels of these osmolytes at all irrigation intervals. In response to drought, plants accumulate osmolytes such as proteins and sugars, which optimize water potential, eliminate ROS, and safeguard cellular components and macromolecules from oxidative damage [57]. Plants produce antioxidants and secondary metabolites in response to abiotic stress, particularly environmental stress (e.g., drought), which play a role in protecting the plant by detoxifying ROS and protecting the plant from these abnormal conditions (i.e., stress), as well as playing a crucial role in protein and amino acids stabilization [58,59]. Under water stress, soluble sugars are a major contingent of compatible solutes that play a key role in reducing the impacts of water stress by modulating turgor pressure or conferring drought opposition to plant cells [60]. Furthermore, soluble sugars act as signaling molecules that control gene expression in plants' stress responses [61]. The increase in total soluble proteins in maize could be attributable to the expression of novel proteins that are involved in the acclimation to water stress. Plants promote the synthesis of proteins such as RAB17 or other proteins involved in glycolysis and the Krebs cycle [62] to eradicate the damage caused by water deficiency. According to Ling et al. [63], late embryogenesis abundant (LEA) proteins and osmotines are also produced to protect cells from dehydration. As a result, the genotypes G10 and G123 have the highest capability to withstand water deficit through promoting cellular protein and sugar synthesis.

Plants' molecular responses to diverse abiotic stressors are known to be related to genome dynamics as a cellular response. Genetic alteration of chromatin is a well-studied stress response in plants [64], and this mechanism affects transcription, cell cycle progression, and DNA repair in response to various environmental stressors [65]. In drought-stressed plants, chromatin organization and remodeling activities through epigenetic mechanisms (DNA methylation and histone modifications) play a critical role in controlling the cell cycle and modifying gene expression [66]. Our results showed that molybdenum cofactor sulfurase (LOS5), checkpoint clamp loader component (Rad17), 9-cis-epoxycarotenoid dioxygenase (NCED1), cationic amino acid transporter 1 (CAT1), and delta1-pyrroline-5-carboxylate synthase 1 (ZmP5CS1) genes were all upregulated in the studied maize genotypes, particularly G10 and G123, following prolonged drought (20 day interval) at the vegetative stage. The gene LOS5 catalyzes the last step in abscisic acid biosynthesis, which is important in plants' response to abiotic stressors [67]. Lu et al. [68] reported that overexpression of the LOS5 encoding gene cloned from *Arabidopsis* substantially improved transgenic maize drought tolerance. The gene RAD17 is a replication factor-like protein, which is required for responses to DNA damage, replication stress, and double-strand break (DSB) repair [69]. RAD17 mutations in *Arabidopsis* demonstrated greater sensitivity to DNA damaging agents as well as a delay in DSB repair [70]. As a result, the overexpression of RAD17 in all maize genotypes, particularly G10 and G123, reflects their drought tolerance potential via its participation in DNA damage repair in response to water stress. Another gene involved in abscisic acid biosynthesis is NCED1, which has been reported to be upregulated under water stress in various plant species [71,72]; NCED1 was reported to be involved in sugar metabolism and abscisic acid accumulation to enhance drought tolerance in water-stressed plants [73]. The gene CAT1 is a stress-responsive gene that promotes cellular repair by delivering amino acids [74]. The overexpression of CAT1 in response to drought stress suggests that it may boost amino acids metabolism in response to water stress [75]. The gene ZmP5CS1 (delta1-pyrroline-5-carboxylate synthase 1) is a proline

synthesis-related gene that can be upregulated by water deficit via an ABA-dependent signaling pathway, resulting in increased proline synthesis [76]. In accordance with our results, Mostafa et al. [77] reported that maize exposure to water deficiency significantly upregulated ZmP5CS1 to promote proline biosynthesis to counter osmotic stress.

## 5. Conclusions

Water stress is a complex process that includes interconnected phenological, physiological, biochemical, and molecular factors. In the current study, morphological, physiological, and molecular differences were investigated in five maize genotypes (G314, G2, G10, G123, and G326) with varying drought tolerance throughout three watering intervals. Drought tolerance is stronger in G10 and G314 genotypes than in other genotypes, possibly due to variations in gene expression linked to drought response. All maize genotypes showed significant stimulation of osmoregulators, antioxidative systems, and water stress-responsive genes, with the superiority of G10 and G123 genotypes. Overall, our findings show that G10 and G123 genotypes can be established in maize fields where prolonged periods of water scarcity are predicted and that they could be used in drought-tolerance breeding programs. More research trials will be required in the future to determine the specific physiological and molecular mechanisms that allow drought-tolerant genotypes to be employed as acceptable plants in water-stressed situations in breading programs.

**Author Contributions:** Conceptualization, K.M.S.-A. and D.G.; methodology, K.M.S.-A., D.G., and A.A.N.; software, K.M.S.-A., D.G., and A.A.N.; validation, K.M.S.-A., D.G., and M.K.H.E.; formal analysis, K.M.S.-A.; investigation, K.M.S.-A., D.G., and A.A.N.; resources, K.M.S.-A., A.A.N., and M.K.H.E.; data curation, K.M.S.-A.; writing—original draft preparation, K.M.S.-A., D.G., A.A.N., and M.K.H.E.; writing—review and editing, K.M.S.-A. and D.G.; visualization, K.M.S.-A., A.A.N., and M.K.H.E.; supervision, K.M.S.-A. and M.K.H.E. All authors have read and agreed to the published version of the manuscript.

**Funding:** This research received no external funding.

**Institutional Review Board Statement:** Not applicable.

**Informed Consent Statement:** Not applicable.

**Data Availability Statement:** Not applicable.

**Conflicts of Interest:** The authors declare no conflict of interest.

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
