# Peer review of "Evaluation of Drought Tolerance of Five Maize Genotypes by Virtue of Physiological and Molecular Responses"

_agronomy, doi:10.3390/agronomy12010059_

Round 1

Reviewer 1 Report

The authors evaluated the effects of drought on five cultivars of Zea mais grown in Egypt in the years 2019-2020. The cultivars were subjected to three different degrees of water stress and some parameters were evaluated including: growth and development, yield in terms of grain production, production of proline and glycinebetaine (osmoregulators), the content of flavonoids and compounds phenolics, catalase activity, total proteins and sugars. Furthermore, the authors evaluated the gene expression in terms of mRNA of genes involved in resistance to drought.

The authors identified different morphological and molecular alterations induced by water stress, and identified the cultivars most resistant to drought.

The manuscript is clear and well written. The authors' hypotheses are well defined and developed through the results. However, I have some suggestions to point out to the authors.

Between these:

Authors should change the title of paragraph 3.1. They should use something more explanatory.

All biochemical and molecular analyzes were performed on the leaves. Because? Why weren't the seeds also used?

Finally, were the cultures in the open? What were the rainfall conditions? What were the soil conditions of the land?

Author Response

Dear Prof. Editor in Chief of Agronomy journal

Firstly, we would like to thank you and your respectful reviewers for the deep insight to our article (Manuscript ID: agronomy-1463220) to be appeared in a suitable form. These comments are all valuable and helpful for improving our article. We have seriously discussed about all these comments. According to the reviewer’s comments, we have tried the best to enhance our manuscript to meet with the requirements of your journal. In this revised version, changes to our manuscript within the document were all highlighted by using track changes button as advised by you and your reviewers. Point-by-point responses to the reviewers are listed below this letter.

If there are any other modifications we could make, we would like very much to modify them and we really appreciate your help. Thank you very much for your help.

Comment

Response

Reviewer #1

1) Authors should change the title of paragraph 3.1. They should use something more explanatory.

The title of the mentioned paragraph (3.1.) was changed according to the reviewer’s recommendation.

2) All biochemical and molecular analyzes were performed on the leaves. Because? Why weren't the seeds also used?

In fact, we would like to thank the reviewer for his valuable comment. We aimed in this study to evaluate the drought resistant genotype among the five studied genotypes based on the growth attributes and physiological and molecular aspects. In a complementary study based on the results of this study, we are now focusing on the yield traits and the metabolomics changes in the yielded seeds.  

3) Finally, were the cultures in the open? What were the rainfall conditions? What were the soil conditions of the land?

The soil conditions for the study's soil have been added to the Materials and Methods section. However, because Egypt is a Mediterranean country, the summer season in Egypt is frequently dry, with no rainfall from May to October.

4) I suggest this manuscript to be included in the manuscript references: Karimmojeni, H.; et al. Competitive Ability Effects of Datura stramonium L. and Xanthium strumarium L. on the Development of Maize (Zea mays) Seeds. Plants 2021, 10, 1922. doi: 10.3390/plants10091922

The suggested reference was added in Materials and Methods section.

Reviewer 2 Report

General comments

I have read the manuscript (agronomy -1463220). Entitle: Evaluation of drought tolerance of five maize genotypes by birtue of physiological and molecular responses. written by Khalil M. Saad-Allah et. al., for publication of agronomy MDPI. In this study, the author investigates metabolic and physiological adjustment mechanisms as well as drought-responsible gene expression pattern of five maize genotypes with varying drought tolerance capacities during the vegetative stage. In this study author did the three different type of treatments (different drought intensity) control, mild drought and heavy drought in two consecutive years. The author found that on the transcription level author found that the prolonged water stress increased the expression of drought –responsive gene. In this study, the relatively water stress tolerant due to improved osmo-regulatory, antioxidant, and metabolic activities under water stress conditions.

The overall research is well conducted and research is obvious application potential for the readers because this research helps to identify the drought resistant maize genotypes under the drought stress conditions. In this sense, the manuscript is much valuable. However, I found some points especially the flow of the text is not smooth and sometimes I found the shallow writing and lack of potential references, and lack of connection of story in different paragraphs especially in the introduction and discussion sections. In discussion, the author should smoother the text which should discussion in-depth of the physiological responses and molecular responses and that will affect plant biology of the different maize genotypes. I also found the lack of potential and appropriate references to support the findings. The author should provide enough examples and their interpretation of different traits of physiological and biochemical responses. I mention some suggestions and recommend some literature. Overall after I evaluate this manuscript, I request this manuscript in “MAJOR REVISION”. I also request to authors for revision according to the rules of the journal and correct the bibliography.

  Major suggestions

 Introduction: Improve your introduction more logically. I saw author did the good starting with the background of the water stress that cause the yield restrictions the agricultural products. Moreover, in the second paragraph of introduction Ln 40 author started to enlist the numerous responses of drought stress but author should focus mainly based on the physiological and molecular perspectives. Author should ideally should shows the physiological responses due to drought stress and its affect the plant biology in the introduction section with referencing many droughts related previous research. Please read and cite this article which are the good references to express the negative effect of drought “Response of drought stress in prunus sargentii and larix kaempferii ...https://doi.org/10.1016/j.foreco.2020.118099” Please mentioned that “drought reduced the morphological and physiological traits, drought reduce the photosynthesis rate, stomatal conductance, leaf water potential, Carbon Isotopic composition and sap movement into the xylem vessel”

Hypothesis of the study: Author well describe the objective of the study. Obviously, the objective is well known that the identify of “drought tolerance genotype”. However, author should still work in the last section of the introduction specially in the hypothesis of the study and its objectives. The Ln no. 91 to 94 is not correct, please rephrase this sentence. Actually, drought tolerance maize genotype helps to cultivate in the water restricted or water limited area and geography only, not need to mention the “water non restricted area” if you mentioned this the scope of yours, is not strength. Objectives of the study is almost okey because the line 85 to 87 may also more clear. But please some clear and modify the text related to the research hypothesis which is comparatively poor in the last section of the study. Hypothesis of the study is important thing and it give another strength for the introduction. The hypothesis should be very clear in the introduction sections because, without appropriate literature, questions, or hypotheses in the introduction section the entire text will be unclear. The author should give special attention and the sequential presentation of the content in the introduction with presenting the hypothesis of the study.

Some others comment

 1) Line no. 96: Author should to be further consider that the 2.1 section. “ material and experimental design”. It is not clear that the watering amount and days of applications. Author should also consider the weather condition. Because it may vary the amount of water and interval of application (10 days). Why author did not consider this in here? Moreover, author start the drought stress from the 21 days, is this a reproductive stage? Not the vegetative stage.

2) Line no. 114: Author should to be consider that the replication of the plant n = ? for measure the shoot biomass, shoot height, leaf area? Please also mention the instrumental detail. How you measure the leaf area? Which instruments, company, replication?

3) Line no. 125: The subtitle, “Osmoregulatory molecules determination” Please concise the text and reduce the unnecessary description. According to the text, author may just refer to the previous literatures if methodology is similar, in this case author should well mentioned the references, (i.e., According to this these scientists we did the similar protocol for measure the…..).

4) Line no. 207: Please make it clear “Two-way ANOVA result for what? In title it should be clear.

5) Line no. 237: Author should be more clearly write the title of the figures or table and legend too, please check this clearly. Author should be modifying the table “3 Title” It is not only enough the “varying irrigation” but also write the heavy drought, light drought or control that make it more clear to the readers.

6) Line no. 261, 270: Author should be applying the above suggestions in the title of table 4 and 5 and … other text of the manuscript where applicable because of the clear meaning. I also suggest to the author that please careful the consistency of the terminology throughout the manuscript.

7) Line no. 466-483 (Discussion): Author should Improve the whole paragraph from line 466 to 483 with the related text of antioxidant and secondary metabolites and specially should to be focus related to the “ROS” rest of the other text author may shift in other paragraph with good flow of writing. The generation of antioxidant and secondary metabolites under the drought stress condition release the ROS (why ROS is emerging in stress condition?) but author lacking some more discussion about these parts (antioxidant and ROS). Refer these two articles read and cited them in the references for better clarify the text specially related to the line no. 470-474. References are (1) https://doi.org/10.1016/j.scitotenv.2021.146466 (2) https://doi.org/10.1038/s41598-019-55889 Please mention somewhere in that paragraph “abiotic stress specially environmental stress (I.e. drought) plant produces the ROS when these plant exposed to the stress condition and plant produce antioxidant, flavonoids, and secondary metabolites play to role for protect the plant for detoxify ROS and protect the plant to protect the abnormal condition (i.e. stress) and protein and amino acid stabilization”.

8) Conclusion section (Line no. 515)

Author should to improve the conclusion because it comparatively weak and repeated the most of the result part. Please change the tone of the presentation specially 521 to 521. In conclusion section not need to describe the methodology parts. Conclusion section should be in good glow with include the gist of the research and should be present the future insight of the research based on your current finding and strength of your results for the future research guideline.

 9) Reference: Line no. 541: please double-check the citations, their style, and spell check, and other grammatical errors. moreover, I request to authors for revision throughout the manuscript according to the journal rules.

Good Luck!

Author Response

Dear Prof. Editor in Chief of Agronomy journal

Firstly, we would like to thank you and your respectful reviewers for the deep insight to our article (Manuscript ID: agronomy-1463220) to be appeared in a suitable form. These comments are all valuable and helpful for improving our article. We have seriously discussed about all these comments. According to the reviewer’s comments, we have tried the best to enhance our manuscript to meet with the requirements of your journal. In this revised version, changes to our manuscript within the document were all highlighted by using track changes button as advised by you and your reviewers. Point-by-point responses to the reviewers are listed below this letter.

If there are any other modifications we could make, we would like very much to modify them and we really appreciate your help. Thank you very much for your help.

Comment

Response

Reviewer #2

1) Introduction: Improve your introduction more logically. I saw author did the good starting with the background of the water stress that cause the yield restrictions the agricultural products. Moreover, in the second paragraph of introduction Ln 40 author started to enlist the numerous responses of drought stress but author should focus mainly based on the physiological and molecular perspectives. Author should ideally shows the physiological responses due to drought stress and its affect the plant biology in the introduction section with referencing many droughts related previous research. Please read and cite this article which are the good references to express the negative effect of drought “Response of drought stress in prunus sargentii and larix kaempferii ...https://doi.org/10.1016/j.foreco.2020.118099” Please mentioned that “drought reduced the morphological and physiological traits, drought reduce the photosynthesis rate, stomatal conductance, leaf water potential, Carbon Isotopic composition and sap movement into the xylem vessel”

We appreciate the reviewer's suggestions for improving our paper to fit the journal's focus. We changed the focus of our introduction to emphasize the physiological and molecular perspectives. We demonstrated the physiological responses to drought stress as well as the impact on plant biology.

2) Hypothesis of the study: Author well describe the objective of the study. Obviously, the objective is well known that the identify of “drought tolerance genotype”. However, author should still work in the last section of the introduction specially in the hypothesis of the study and its objectives. The Ln no. 91 to 94 is not correct, please rephrase this sentence. Actually, drought tolerance maize genotype helps to cultivate in the water restricted or water limited area and geography only, not need to mention the “water non restricted area” if you mentioned this the scope of yours, is not strength. Objectives of the study is almost okey because the line 85 to 87 may also more clear. But please some clear and modify the text related to the research hypothesis which is comparatively poor in the last section of the study. Hypothesis of the study is important thing and it give another strength for the introduction. The hypothesis should be very clear in the introduction sections because, without appropriate literature, questions, or hypotheses in the introduction section the entire text will be unclear. The author should give special attention and the sequential presentation of the content in the introduction with presenting the hypothesis of the study.

The objective of the study was corrected in the last section of the introduction as suggested by the reviewer.

3) Line no. 96: Author should to be further consider that the 2.1 section. “ material and experimental design”. It is not clear that the watering amount and days of applications. Author should also consider the weather condition. Because it may vary the amount of water and interval of application (10 days). Why author did not consider this in here? Moreover, author start the drought stress from the 21 days, is this a reproductive stage? Not the vegetative stage.

In the Materials and Methods section, the watering system and amount of water used were clearly stated. The experimental site's weather conditions were also mentioned.

The use of different water regimes had started at the beginning of the vegetative stage of growth (after 21 days of seed sowing).

4) Line no. 114: Author should to be consider that the replication of the plant n = ? for measure the shoot biomass, shoot height, leaf area? Please also mention the instrumental detail. How you measure the leaf area? Which instruments, company, replication?

All the details of growth parameters including shoot biomass and leaf area were mentioned in the Materials and Methods section.

5) Line no. 125: The subtitle, “Osmoregulatory molecules determination” Please concise the text and reduce the unnecessary description. According to the text, author may just refer to the previous literatures if methodology is similar, in this case author should well mentioned the references, (i.e., According to this these scientists we did the similar protocol for measure the…..).

The osmoregulatory molecules determination was summarized by reducing the unnecessary description as recommended by the reviewer.

6) Line no. 207: Please make it clear “Two-way ANOVA result for what? In title it should be clear.

The title of the mentioned paragraph was changed according to the reviewer’s recommendation to fulfil the experiment description.

7) Line no. 237: Author should be more clearly write the title of the figures or table and legend too, please check this clearly. Author should be modifying the table “3 Title” It is not only enough the “varying irrigation” but also write the heavy drought, light drought or control that make it more clear to the readers.

The ligand of tables was rewritten in the way recommended by the reviewer to reflect the experimental design clearly, giving more clear vision to the readers.

8) Line no. 261, 270: Author should be applying the above suggestions in the title of table 4 and 5 and … other text of the manuscript where applicable because of the clear meaning. I also suggest to the author that please careful the consistency of the terminology throughout the manuscript.

The above suggestions was applied in the title of tables 4, 5 6, 7, and 8 as recommended. Also, the consistency of the terminology throughout the manuscript was considered.

9) Line no. 466-483 (Discussion): Author should Improve the whole paragraph from line 466 to 483 with the related text of antioxidant and secondary metabolites and specially should to be focus related to the “ROS” rest of the other text author may shift in other paragraph with good flow of writing. The generation of antioxidant and secondary metabolites under the drought stress condition release the ROS (why ROS is emerging in stress condition?) but author lacking some more discussion about these parts (antioxidant and ROS). Refer these two articles read and cited them in the references for better clarify the text specially related to the line no. 470-474. References are (1) https://doi.org/10.1016/j.scitotenv.2021.146466 (2) https://doi.org/10.1038/s41598-019-55889. Please mention somewhere in that paragraph “abiotic stress specially environmental stress (I.e. drought) plant produces the ROS when these plant exposed to the stress condition and plant produce antioxidant, flavonoids, and secondary metabolites play to role for protect the plant for detoxify ROS and protect the plant to protect the abnormal condition (i.e. stress) and protein and amino acid stabilization”.

We appreciate the reviewer's proposal, which would improve the Discussion section. To maintain a good flow of writing, the indicated paragraph was revised and given a proper citation.

10) Conclusion section (Line no. 515)

Author should to improve the conclusion because it comparatively weak and repeated the most of the result part. Please change the tone of the presentation specially 521 to 521. In conclusion section not need to describe the methodology parts. Conclusion section should be in good glow with include the gist of the research and should be present the future insight of the research based on your current finding and strength of your results for the future research guideline.

The conclusion section was enhanced by removing the redundant parts, as suggested by the reviewer. In addition, future research insights based on our results for the study guideline were provided.

11) Reference: Line no. 541: please double-check the citations, their style, and spell check, and other grammatical errors. moreover, I request to authors for revision throughout the manuscript according to the journal rules.

Thank you for the reviewer's advice, which enabled our paper to be submitted in a format that complied with the journal's guidelines. The citations and the entire manuscript were double-checked for grammatical and spelling mistakes.

Reviewer 3 Report

Dear authors, here are the most important points which should be corrected:

Materials and methods are poorly written, which does not allow to repeat such experiments and makes it difficult to review the results.

Several things are missing:

  1. The experiment was conducted for two years (2019 and 2020), some climatic conditions at the experimental site during these vegetations should be presented. This was an experiment with different irrigation schedules; thus, it is important to state if there was precipitation during the vegetation period, which were the temperatures, evapotranspiration could be calculated etc.
  2. There is no data about the soil type and chemical and physical soil properties. Plant available water is determined by soil texture and structure. Thus, it is critical to give detailed information about the soil in such experiments. Data about soil field capacity and wilting point should be presented as well as the soil water content at each irrigation treatment.
  3. Information about irrigation is missing. How much water was applied during each irrigation, which type of irrigation was used etc.
  4. There is no information about the number of samples for the biochemical analysis (CAT, POD, proline, GB, phenolics, flavonoids etc.)
  5. The experimental design was split-plot; it should be indicated what was the main plot and what was the subplot.
  6. The molecular analysis is not clear if it was performed on all three irrigation treatments because, in the result section, it only shows the differences among maize genotypes.
  7. The experiment was conducted for two years, and different year is not included in the statistical analysis. There are probably some differences between these two years.

Introduction

Some literature review should be included in the introduction about the molecular mechanisms related to drought in maize.

Results

Since there is significant genotype x irrigation treatment interaction for most of the studied traits (Table 2).  Authors should show interactions (pairwise differences) and not only the differences between treatments. It would be of interest to see which genotype perform better under reduced irrigation.

Authors should consistently use term control irrigation treatment throughout the manuscript and avoid terms such as typical irrigation treatment, normal irrigation treatment, or routinely irrigated plants.

Author Response

Dear Prof. Editor in Chief of Agronomy journal

Firstly, we would like to thank you and your respectful reviewers for the deep insight to our article (Manuscript ID: agronomy-1463220) to be appeared in a suitable form. These comments are all valuable and helpful for improving our article. We have seriously discussed about all these comments. According to the reviewer’s comments, we have tried the best to enhance our manuscript to meet with the requirements of your journal. In this revised version, changes to our manuscript within the document were all highlighted by using track changes button as advised by you and your reviewers. Point-by-point responses to the reviewers are listed below this letter.

If there are any other modifications we could make, we would like very much to modify them and we really appreciate your help. Thank you very much for your help.

Comment

Response

Reviewer #3

1) The experiment was conducted for two years (2019 and 2020), some climatic conditions at the experimental site during these vegetations should be presented. This was an experiment with different irrigation schedules; thus, it is important to state if there was precipitation during the vegetation period, which were the temperatures, evapotranspiration could be calculated etc.

Thank you for contributing to the improvement of our experimental design. The climatic conditions at the experimental site, precipitation, temperature and relative humidity were added in the Materials and Methods section.

2) There is no data about the soil type and chemical and physical soil properties. Plant available water is determined by soil texture and structure. Thus, it is critical to give detailed information about the soil in such experiments. Data about soil field capacity and wilting point should be presented as well as the soil water content at each irrigation treatment.

The chemical and physical properties and the soil type of the experimental site, as well as the field capacity of the soil and  the amount of water added at each irrigation treatment were added to the experimental design section in the Materials and Methods part.

3) Information about irrigation is missing. How much water was applied during each irrigation, which type of irrigation was used etc.

The amount of water that was applied during each irrigation (261 L/m2) was added in the experimental design part.

4) There is no information about the number of samples for the biochemical analysis (CAT, POD, proline, GB, phenolics, flavonoids etc.)

The information about the number of samples used in the biochemical analysis are provided in the statistical analysis. The lastest sentence in this part state that “The results are reported as the average of two separate studies with six replicates (3 each) ± standard deviation”.

5) The experimental design was split-plot; it should be indicated what was the main plot and what was the subplot.

In the experimental design section the main and sub plots were indicated by adding the sentence “The main-plot was allocated irrigation interval treatments, while maize genotypes were assigned to the sub-plots, with three replications for each”.

6) The molecular analysis is not clear if it was performed on all three irrigation treatments because, in the result section, it only shows the differences among maize genotypes.

The molecular analysis was performed on the longest irrigation interval (20 days) on all maize genotypes. This was referred in the results section by the sentence “Prolonged irrigation interval (20 days) at the vegetative stage significantly affected the expression levels of some genes related to water stress tolerance in maize, such as molybdenum cofactor sulfurase (LOS5), checkpoint clamp loader component (Rad17), 9-cis-epoxycarotenoid dioxygenase (NCED1), cationic amino acid transporter 1 (CAT1), and delta1-pyrroline-5-carboxylate synthase 1 (ZmP5CS1) (Table 9).”

7) The experiment was conducted for two years, and different year is not included in the statistical analysis. There are probably some differences between these two years.

We appreciate the reviewer criticism in this point of view. However, throughout the statistical analysis tests, we integrated the findings of the two years since we noticed that they were substantially similar. Thus we added at the end of Materials and Methods section that the results are reported as the average of two separate studies with six replicates (3 each) ± standard deviation.

8) Some literature review should be included in the introduction about the molecular mechanisms related to drought in maize.

The introduction section was provided with some literature about the molecular mechanisms related to drought in maize.

10) Authors should consistently use term control irrigation treatment throughout the manuscript and avoid terms such as typical irrigation treatment, normal irrigation treatment, or routinely irrigated plants.

As proposed by the reviewer, the term "control irrigation treatment" was used throughout the manuscript in place of "typical irrigation treatment," "standard irrigation treatment," or "routinely watered plants."

Round 2

Reviewer 2 Report

Dear Author

I have read the revised manuscript (agronomy-1463220). Entitle: Evaluation of drought tolerance of five maize genotypes by virtue of physiological and molecular responses written by Khalil M et. al., for publication of agronomy MDPI. Author addressed all the questions and suggestions what I raised issue in the review of the original manuscript. I satisfy the author revisions throughout the paper. The abstract issue is well written by the author. Now this manuscript improved the flow of the flow of writing, which was comparatively shallow in the original version but in this revised copy author address all the quarries and suggestions where the introduction is significantly improved by author. Before accept this manuscript, I request to author to check the whole manuscript by native speaker for correct spell check, and other grammatical errors.

Author Response

Dear Prof. Editor in Chief of Agronomy journal

Firstly, we would like to thank you and your respectful reviewers for the deep insight to our article (Manuscript ID: agronomy-1463220) to be appeared in a suitable form. These comments are all valuable and helpful for improving our article. We have seriously discussed about all these comments. According to the reviewer’s comments, we have tried the best to enhance our manuscript in the second round of revision to meet with the requirements of your journal. In this revised version (R2), changes to our manuscript within the document were all highlighted by using track changes button as advised by you and your reviewers. Point-by-point responses to the reviewers are listed below this letter.

If there are any other modifications we could make, we would like very much to modify them and we really appreciate your help. Thank you very much for your help.

Comment

Response

Reviewer #2

1) I have read the revised manuscript (agronomy-1463220). Entitle: Evaluation of drought tolerance of five maize genotypes by virtue of physiological and molecular responses written by Khalil M et. al., for publication of agronomy MDPI. Author addressed all the questions and suggestions what I raised issue in the review of the original manuscript. I satisfy the author revisions throughout the paper. The abstract issue is well written by the author. Now this manuscript improved the flow of the flow of writing, which was comparatively shallow in the original version but in this revised copy author address all the quarries and suggestions where the introduction is significantly improved by author. Before accept this manuscript, I request to author to check the whole manuscript by native speaker for correct spell check, and other grammatical errors.

The reviewer's suggestion to check the entire document for possible spelling and grammatical problems is one we welcome. We've already sent the paper to an English colleague at Bath University to double-check spelling and grammar. The corrections made to our article can be seen in the revised version (R2) of our manuscript.

Reviewer 3 Report

LINES 52-56:  I'm sure that you can find such information about maize. Please find references which talk about maize.

287-297. Statistical analysis is still not clear enough

  • please include a sentence about the effect of the different years (explain, why you haven't used different years in the model).
  • LINE 295-297 Please explain/correct this.

„The differences in growth performance, yield attributes, osmoregulatory molecules, phenols and flavonoids, CAT and POD activities, soluble proteins and sugars, and gene expression data between water stress treated and control maize genotypes were assessed using one-way analysis of variance (One-way ANOVA) followed by Tukey's HSD test at the 5% level for mean comparison.“

First, you have performed Two-way ANOVA (table 2), and then you have used ON-way ANOVA and post hoc Tukey test for testing the difference among irrigation treatments for each genotype?

However, you are presenting all results (growth performance, yield attributes, osmoregulatory molecules, phenols and flavonoids, CAT and POD activities, soluble proteins and sugars, except the gene expression) as you have performed two-way ANOVA? Tables are shown in this way, see tables 3-8. But you have tested only the differences among irrigation treatments for each genotype (One-way ANOVA, post hoc Tukey comparisons). This is wrong.

In your experiment, you have two years, five genotypes and three irrigation levels. In addition, your aims are: „The objective of this study was to use morphological, physiological, and molecular data to investigate the variations in water-stress tolerance of five maize genotypes, commonly grown in Egypt, under three different water regimes. We also aimed to detect the most tolerant genotype that could be used in water-restricted areas that might be employed as parents for water-stress-tolerant genotypes.”

If you neglect the effect of year and state that those two years were very similar,  you should still test and compare the effects of genotype and irrigation treatments (if you have significant interaction). However, in the results, you only the differences among irrigation treatments. You can not discuss and make conclusions about genotype performance under the different irrigation regimes if you neglect the significant interaction genotype x irrigation treatment (see Table 2).

If you have significant interaction, after Two-way ANOVA, you should use pairwise difference Tukey’s test.  In such a way you can compare the performance of different genotypes under different irrigation levels, and find which one is the best.

Author Response

Dear Prof. Editor in Chief of Agronomy journal

Firstly, we would like to thank you and your respectful reviewers for the deep insight to our article (Manuscript ID: agronomy-1463220) to be appeared in a suitable form. These comments are all valuable and helpful for improving our article. We have seriously discussed about all these comments. According to the reviewer’s comments, we have tried the best to enhance our manuscript in the second round of revision to meet with the requirements of your journal. In this revised version (R2), changes to our manuscript within the document were all highlighted by using track changes button as advised by you and your reviewers. Point-by-point responses to the reviewers are listed below this letter.

If there are any other modifications we could make, we would like very much to modify them and we really appreciate your help. Thank you very much for your help.

Comment

Response

Reviewer #3

1) LINES 52-56:  I'm sure that you can find such information about maize. Please find references which talk about maize.

Thank you for suggesting that we include a citation to document our talk about maize. In our amended version of the manuscript, we've added three references to support our talk on maize.

2) L. 287-297: Statistical analysis is still not clear enough. Please include a sentence about the effect of the different years (explain, why you haven't used different years in the model).

We explained the cause of not include the different years in the statistical analysis by adding the following sentence “All of the experimental results were merged and analyzed over two years, because it was assumed that genotype effects were constant and year effects were random.”

3) L. 295-297: Please explain/correct this. The differences in growth performance, yield attributes, osmoregulatory molecules, phenols and flavonoids, CAT and POD activities, soluble proteins and sugars, and gene expression data between water stress treated and control maize genotypes were assessed using one-way analysis of variance (One-way ANOVA) followed by Tukey's HSD test at the 5% level for mean comparison.“

The sentence in question was changed as suggested.

First, you have performed Two-way ANOVA (table 2), and then you have used ON-way ANOVA and post hoc Tukey test for testing the difference among irrigation treatments for each genotype?

However, you are presenting all results (growth performance, yield attributes, osmoregulatory molecules, phenols and flavonoids, CAT and POD activities, soluble proteins and sugars, except the gene expression) as you have performed two-way ANOVA? Tables are shown in this way, see tables 3-8. But you have tested only the differences among irrigation treatments for each genotype (One-way ANOVA, post hoc Tukey comparisons). This is wrong.

In your experiment, you have two years, five genotypes and three irrigation levels. In addition, your aims are: „The objective of this study was to use morphological, physiological, and molecular data to investigate the variations in water-stress tolerance of five maize genotypes, commonly grown in Egypt, under three different water regimes. We also aimed to detect the most tolerant genotype that could be used in water-restricted areas that might be employed as parents for water-stress-tolerant genotypes.”

If you neglect the effect of year and state that those two years were very similar,  you should still test and compare the effects of genotype and irrigation treatments (if you have significant interaction). However, in the results, you only the differences among irrigation treatments. You can not discuss and make conclusions about genotype performance under the different irrigation regimes if you neglect the significant interaction genotype x irrigation treatment (see Table 2).

If you have significant interaction, after Two-way ANOVA, you should use pairwise difference Tukey’s test.  In such a way you can compare the performance of different genotypes under different irrigation levels, and find which one is the best.

First and foremost, we would like to express our gratitude for your recommendation to reconsider the statistical analysis. In reality, it was a good point that gave our experiment purpose. Of course, you have complete authority to demand that we pay attention to that argument. To investigate the difference in genotype performance under varied watering intervals, we redesigned two-way ANOVA to tables 3-8.

We used pairwise difference Tukey's test after the two-way ANOVA to determine the significance of the used interactions. We are confident that the update we made to the paper will allow us to evaluate the performance of different genotypes under various watering intervals and determine which genotype performed best under those situations.